# 4DEditPro: Progressively Editing 4D Scenes from Monocular Videos with Text Prompts

## Abstract

Editing 4D scenes using text prompts is a novel task made possible by advances in text-to-image diffusion models and differentiable scene representations. However, conventional approaches typically use multi-view images or videos with camera poses as input, which causes inconsistencies when editing monocular videos due to the reliance of these tools on iteratively per-image editing and the absence of multi-view supervision. Furthermore, these techniques usually require external Structure-from-Motion (SfM) libraries for camera pose estimation, which can be impractical for casual monocular videos. To tackle these hurdles, we present **4DEditPro**, a novel framework that enables consistent 4D scene editing on casual monocular videos with text prompts. In our 4DEditPro, the Temporally Propagated Editing (TPE) module guides the diffusion model to ensure temporal coherence across all input frames in scene editing. Furthermore, the Spatially Propagated Editing (SPE) module in 4DEditPro introduces auxiliary novel views near the camera trajectory to enhance the spatial consistency of edited scenes. 4DEditPro employs a pose-free 4D Gaussian Splatting (4DGS) approach for reconstructing dynamic scenes on monocular videos, which progressively recovers relative camera poses, reconstructs the scene, and facilitates scene editing. We have conducted extensive experiments to demonstrate the effectiveness of our approach, including both quantitative measures and user studies.

## 1 Introduction

In recent years, notable progress has been made in differentiable scene representations from multi-view images (Mildenhall et al., 2021; Pumarola et al., 2021; Kerbl et al., 2023; Wu et al., 2024) as well as text-to-image (T2I) diffusion models (Rombach et al., 2022; Hertz et al., 2023; Brooks et al., 2023; Zhang et al., 2023). By integrating these two lines of research, a variety of approaches (Poole et al., 2023; Wang & Shi, 2023; Park et al., 2024; Cheng et al., 2024) have been proposed to facilitate the generation and editing of 3D contents from text or multi-view images, demonstrating great potential for various applications such as VR/AR and the MetaVerse. Some methods (Shao et al., 2023; Mou et al., 2024) has taken a step further to explore the editing of 4D dynamic scenes, and Control4D (Shao et al., 2023) has leveraged Generative Adversarial Networks (GANs) (Goodfellow et al., 2020) to support diffusion models in producing consistent outcomes from editing. Furthermore, Instruct 4D-to-4D (Mou et al., 2024) treats 4D scenes as pseudo-3D scenes and employs a video editing approach to iteratively generate coherent edited datasets.

Despite these significant advancements, 4D scene editing using only casual videos (i.e., monocular videos with unknown camera poses) remains relatively under-explored, and directly integrating T2I diffusion models with differentiable 4D representations presents several challenges. Firstly, maintaining temporal and spatial consistency is essential for high-quality 4D editing. Previous work (Haque et al., 2023; Shao et al., 2023) has relied on iteratively updating the edited scene until convergence. Yet, the absence of multi-view information in monocular videos significantly reduces the coherence of the edited scene. Secondly, existing 4D editing approaches heavily depend on camera pose estimation techniques (e.g., COLMAP (Schonberger & Frahm, 2016)), which not only introduce redundancy during model initialization, but also cause 4D editing not directly applicable to casual monocular video input.

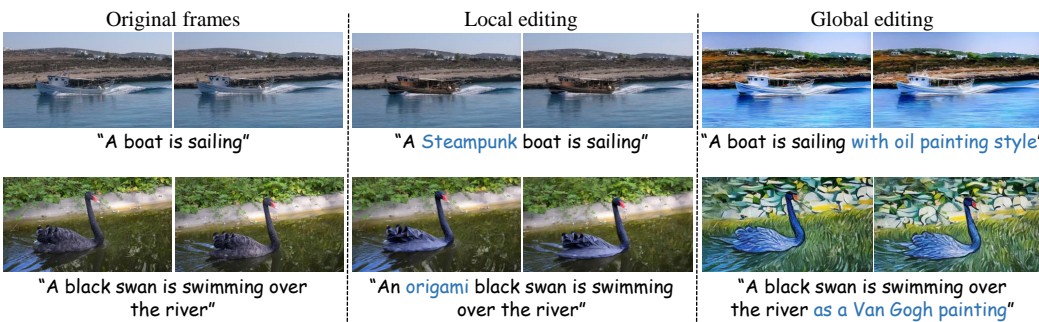

Figure 1: **Results of 4DEditPro**. Using text prompts, our 4DEditPro can generate 4D consistent editing results from monocular videos on local and global editing. Notably, our method does not rely on COLMAP (Schonberger & Frahm, 2016) for pose calculation.

In this paper, we address these challenges in 4DEditPro, our framework that integrates diffusion models and 4D Gaussian Splatting (4DGS) for text prompt-guided 4D scene editing of casual monocular videos. 4DEditPro adapts two key techniques to ensure the temporal and spatial consistency of the editing results. (i) We present the Temporally Propagated Editing (TPE) module to enhance temporal editing coherence. Specifically, we convert all video frames into latent tokens using DDIM inversion (Song et al., 2021) and partition the latent tokens into several batches. Within each batch, we select a reference token. Utilizing the diffusion model with an extended self-attention module based on the text prompt, we extract attention features from reference tokens and propagate these features to align with those of the remaining tokens, ensuring a consistent editing style throughout the entire sequence. (ii) We introduce the Spatially Propagated Editing (SPE) module to mitigate the lack of multi-view information in monocular videos. First, we interpolate novel views near the original camera trajectory. Similar to TPE, we propagate attention features from known views to novel views at a given timestamp, refining the visual quality of novel views.

Moreover, we introduce a progressive dynamic representation utilizing 4DGS for its efficiency in training and rendering. Specifically, we introduce local 3DGS to progressively capture relative pose changes for updating the current pose. Furthermore, we model the global 4DGS with time-dependent opacity, position, and rotation, incorporating the camera pose input from the local 3DGS. We conduct extensive experiments on a diverse set of monocular videos without camera pose input, employing various text prompts to demonstrate the efficacy of our approach. As illustrated in Fig. 1, we utilize both local editing (*e.g.*, editing specific objects or regions) and global editing (*e.g.*, applying a style or weather to the scene) to present the editing ability of our approach. The evaluation demonstrates the effectiveness of our method in producing high-quality rendering results and maintaining temporal-spatial consistency in 4D editing.

In summary, our primary contributions can be outlined as follows:

- We present 4DEditPro, a novel framework that facilitates 4D scene editing from casual monocular videos using text prompts.
- We propose Temporally Propagated Editing (TPE) and Spatially Propagated Editing (SPE) based on diffusion models to achieve temporal and spatial consistency in 4D editing.
- We develop a progressive 4D Gaussian Splatting to accurately and efficiently model scene attributes without requiring camera pose input.
- Extensive experiments on a range of 4D scenes demonstrate the fidelity and consistency of our method in global and local editing.

## 2 BACKGROUND AND RELATED WORK

### 2.1 PRELIMINARY

**3D Gaussian Splatting.** Gaussian Splatting (Kerbl et al., 2023) is an explicit point-based 3D representation. Unlike implicit 3D representations (Mildenhall et al., 2021; Wang et al., 2021), which

generate images through volume rendering, 3DGS adopts a splatting technique for image rendering by projecting a set of 3D Gaussians onto 2D planes. Each *Gaussian ellipse* is characterized by a color $c$ represented with spherical harmonics coefficients, an opacity $o$, a position center $\mu$, and a *covariance matrix* $\Sigma$. The Gaussian ellipse can be calculated as $G(x) = e^{-\frac{1}{2}x^T\Sigma^{-1}x}$, where $x$ represents the displacement from the center $\mu$. The covariance matrix $\Sigma$ can be decomposed into a *rotation matrix* $R$ and a *scaling matrix* $S$ to facilitate differentiable optimization: $\Sigma = RSS^T R^T$. During the projection of 3D Gaussians for rendering onto 2D planes, the *splatting* operation (Zwicker et al., 2001) is employed to position the Gaussians, involving a new covariance matrix $\Sigma'$ in camera coordinates defined as $\Sigma' = JW\Sigma W^T J^T$, where $J$ represents the Jacobian of the affine approximation of the projective transformation, and $W$ denotes a given viewing transformation matrix. The rendering result $C$ at a pixel is achieved by approximating the projection of a 3D Gaussian along the depth dimension onto the pixel: $C = \sum_{i \in N} c_i o_i \prod_{j=1}^{i-1}(1 - o_j)$, where $N$ is the series of ordered points that project onto the pixel, ensuring a coherent rendering of overlapping Gaussians.

**Diffusion-based Editing.** Stable Diffusion (SD) (Rombach et al., 2022) is a leading text-to-image diffusion model that operates within a latent image space. SD encodes RGB images into the latent image space and utilizes a decoder to reconstruct the latent representations into high-resolution images. The core of SD is based on a U-Net architecture (Ronneberger et al., 2015) incorporating residual, self-attention, and cross-attention blocks. Building upon SD, several diffusion models that integrate additional U-Net encoders have been developed (Zhang et al., 2023; Brooks et al., 2023). These U-Net encoders enable image generation controlled by various types of information, such as depth, edges, or specific regions based on prompts. The majority of current 3D or 4D scene editing methods (Haque et al., 2023; Mou et al., 2024) utilize 2D diffusion models with given prompts to edit datasets. These edited datasets are then utilized as training targets to reconstruct the 3D scene.

## 2.2 RELATED WORK

**4D Neural Scene Representation.** Neural representations (Sitzmann et al., 2019; Aliev et al., 2020; Thies et al., 2019) have been applied in various 3D tasks, with Neural Radiance Fields (NeRF) (Mildenhall et al., 2021) being a groundbreaking technique that utilizes volume rendering to optimize 3D modeling with only 2D supervision. However, the rendering process in NeRF is time-consuming. Recently, 3D Gaussian Splatting (3DGS) (Kerbl et al., 2023) has shown impressive rendering quality and speed in 3D reconstruction. The efficient differentiable rendering implementation and explicit representation of 3DGS allow fast training, making it widely used for 4D reconstruction and generation tasks (Yang et al., 2024; Katsumata et al., 2023; Wu et al., 2024; Gao et al., 2024). However, 3DGS typically requires a point cloud generated by COLMAP (Schonberger & Frahm, 2016) for estimation, making it not directly applicable to reconstructing casual videos, thus adding additional steps for downstream tasks. Recent progress has been seen in pose-free 4D Gaussian representations (Wang et al., 2024; Chu et al., 2024; Li et al., 2024), but these representations rely on deformable networks to model time-varying parameters, resulting in extra computational cost for generation or editing tasks. In our approach, we propose a progressive and efficient 4D Gaussian Splatting framework that does not rely on COLMAP initialization, simplifying and optimizing the 4D editing process.

**Diffusion-Based Scene Editing.** Diffusion models iteratively transform random samples into data resembling the target data (Song et al., 2021; Dhariwal & Nichol, 2021), as widely used in various generation tasks such as text-to-image generation or editing (Meng et al., 2022; Couairon et al., 2022). However, in 3D and 4D scene reconstruction and editing, directly applying these models has issues on spatial and temporal consistency. To address these issues, recent diffusion-based 3D editing techniques (Kamata et al., 2023; Haque et al., 2023; Dong & Wang, 2024; Chen et al., 2024; Yu & Liu, 2024) utilize self-distillation as a 2D prior to modify scene appearance, yielding impressive results. Most of these editing approaches incorporate InstructPix2Pix (IP2P) (Brooks et al., 2023), an image-conditioned diffusion model, for instruction-based 2D image editing. For 4D scene editing, Control4D (Shao et al., 2023) proposes to construct a continuous 4D space by training a 4D GAN (Goodfellow et al., 2020) from ControlNet (Zhang et al., 2023) to address inconsistent supervision signals in 4D portrait editing. In comparison, Instruct 4D-to-4D (Mou et al., 2024) regards 4D scenes as pseudo-3D scenes and utilizes warping to propagate editing outcomes. As such, Instruct 4D-to-4D relies heavily on the accuracy of optical flow and may result in some artifacts. Furthermore, these 4D methods may struggle to reconstruct and edit 4D scenes from sparse views or monocular videos, potentially overlooking consistency in both temporal and spatial views.

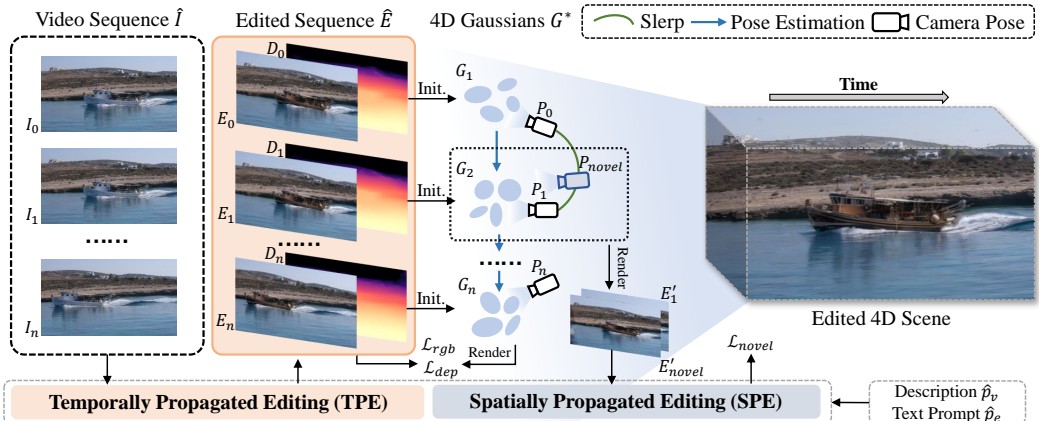

Figure 2: **Our proposed 4DEditPro.** This pipeline utilizes the TPE module to generate a temporally consistent video sequence, employs the SPE module to interpolate and refine novel views near the camera trajectory of the original monocular video (Sec.3.1), and integrates a progressive 4D Gaussian representation for estimating camera poses and reconstructing the 4D scenes (Sec.3.2).

## 3 METHOD

We define the task of 4D scene editing on Gaussian Splatting from casual monocular videos as follows: Given a video sequence $\hat{I}$, a text description $\hat{p}_v$ of the video, and a text prompt $\hat{p}_e$ describing the target editing, generate a 4D scene based on $\hat{I}$, aligning with the text prompt $\hat{p}_e$, and maintaining temporal and spatial consistency. As illustrated in Fig. 2, we employ the TPE module to generate a temporally consistent sequence $\hat{E}$ from the input video sequence $\hat{I}$ with $\hat{p}_v$ and $\hat{p}_e$. Subsequently, we progressively reconstruct the 4D scene based on 4DGS. To improve the spatial consistency of the 4D scene, we introduce the SPE module to enhance the visual quality of novel views near the original camera trajectory. In the following, we describe the key components of our pipeline.

### 3.1 TEMPORALLY-SPATIALLY PROPAGATED EDITING

Previous studies on 4D editing (Mou et al., 2024; Shao et al., 2023) have incorporated 3D warping or GAN assistance to achieve consistent editing results. However, when dealing with monocular videos that lack multi-view information, these methods struggle to accurately edit corresponding regions across frames. This limitation often leads to visual artifacts and unstable editing outcomes, reducing the editing performance.

To address these challenges, we draw inspiration from recent studies (Geyer et al., 2024; Liu et al., 2024) and propose a novel approach called Temporally Propagated Editing (TPE), leveraging latent tokens from DDIM inversion to ensure editing consistency across all frames in monocular videos. Also, we introduce Spatially Propagated Editing (SPE) to refine 4D scene reconstruction by enhancing spatial editing from known views to novel views near the camera trajectory. Our approach aims to enhance both temporal and spatial editing consistency, boosting the overall performance of 4D scene reconstruction.

**DDIM Inversion.** Given an input video sequence consisting of $n$ frames $\hat{I} = [I_1, I_2, ..., I_n]$ and its associated description $\hat{p}_v$, we utilize DDIM inversion on each frame to extract the latent tokens $\phi(z^t)$. This process involves applying a pretrained and fixed text-to-image diffusion model $\varepsilon$ as follows:

$$\epsilon^t = \varepsilon_u(z^t, t, T(\hat{p}_v)),$$

$$\phi(z^t) = \sqrt{\sigma^t} \cdot \frac{z^t - \sqrt{1 - \sigma^{t-1}} \cdot \epsilon^t}{\sqrt{\sigma^{t-1}}} + \sqrt{1 - \sigma^t} \cdot \epsilon^t, \tag{1}$$

where $\varepsilon_u$ represents the U-Net component in the diffusion model $\varepsilon$, and $T(\cdot)$ denotes the text encoder. The variable $t$ corresponds to the timestep of the diffusion process, and $\sigma^t$ represents the scheduling coefficient in the DDIM scheduler.

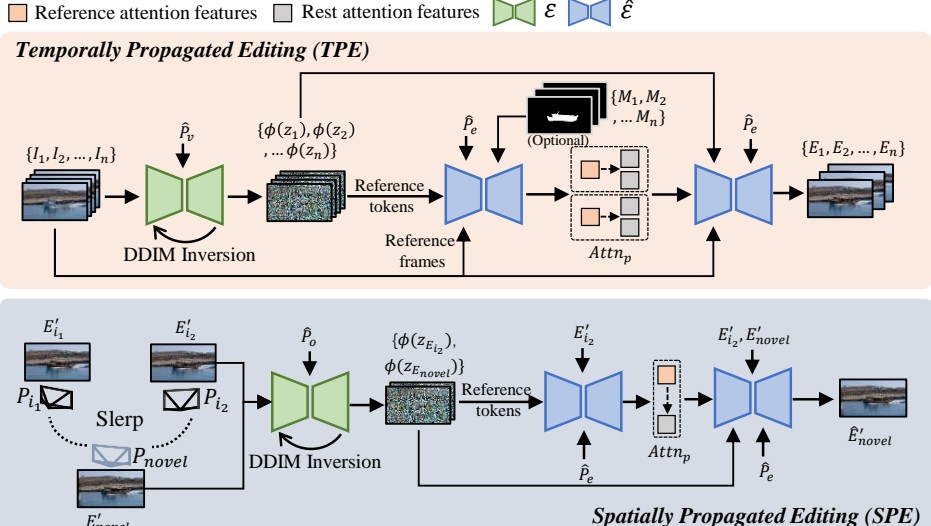

Figure 3: Our proposed TPE and SPE modules. In this example, $\hat{p}_v$, $\hat{p}_e$, and $\hat{p}_o$ correspond to "A boat is sailing", "A Steampunk boat is sailing" and an empty prompt, respectively, with $\varepsilon$ and $\hat{\varepsilon}$ representing the diffusion model and the diffusion model with the extended self-attention block.

**Temporally Propagated Editing.** After obtaining the latent tokens through DDIM inversions, we utilize these tokens to further establish temporal consistency. To optimize GPU memory usage and editing efficiency, we divide the entire set of latent tokens into batches and select one reference latent token within each batch (see Appendix for details). Next, we extract reference attention features from these reference tokens utilizing the diffusion models with the extended self-attention module and then propagate these reference attention features to align with those of the remaining tokens, ensuring consistent editing outcomes.

As illustrated in Fig. 3, we select a reference token from each batch and perform joint editing on these reference tokens with the text prompt $\hat{p}_e$ using the U-Net $\hat{\varepsilon}_u$ of diffusion models that incorporates the extended self-attention block (Wu et al., 2023) to extract reference attention features within reference tokens $\phi(z_{r_n})$:

$$\text{Attn}_{r_1, r_2} = \text{Softmax}(\frac{Q(\phi(z_{r_1}))K(\phi(z_{r_2}))}{\sqrt{d}})V(\phi(z_{r_2})), \tag{2}$$

where $Q(\cdot)$, $K(\cdot)$, and $V(\cdot)$ are linear projections used to acquire Query, Key, and Value features from the self-attention mechanism, with $d$ serving as a scaling factor.

Subsequently, we propagate these reference attention features from $\hat{\varepsilon}$ to their corresponding batches, aligning reference attention features with rest attention features of the non-reference latent tokens to maintain a consistent editing style across the entire temporal sequence:

$$\text{Attn}_p = \omega \cdot \text{Attn}_{p,p} + (1 - \omega) \cdot \frac{1}{N_p} \sum_{i=1}^{N_p} \text{Attn}_{p,r}, \tag{3}$$

where $\omega \in [0, 1]$. This propagation ensures temporal consistency across all frames. Subsequently, we acquire the edited latent $\phi(z_e^0)$ via the denoising process:

$$\epsilon^t = \hat{\varepsilon}_u(\phi(z^t), t, T(\hat{p}_o)) + \omega_u \cdot (\hat{\varepsilon}_u(\phi(z^t), t, T(\hat{p}_e)) - \hat{\varepsilon}_u(\phi(z^t), t, T(\hat{p}_o)),$$

$$\phi(z_e^{t-1}) = \sqrt{\sigma^{t-1}} \cdot \frac{\phi(z_e^t) - \sqrt{1 - \sigma^t} \cdot \epsilon^t}{\sqrt{\sigma^t}} + \sqrt{1 - \sigma^{t-1}} \cdot \epsilon^t, \tag{4}$$

In Eq. 4, $\hat{p}_o$ represents the empty prompt, and $\omega_u$ denotes the classifier-free guidance (Ho & Salimans, 2022). The VAE decoder within the diffusion model is utilized to decode $\phi(z_e^0)$ and generate the final edited images $\hat{E} = [E_1, E_2, .., E_n]$.

To enhance local editing, we incorporate the Lang SAM method (Kirillov et al., 2023) to extract specific masks $m$. These masks are then utilized to confine the editing regions during the denoising process.

$$\phi(z_e^t) = \phi(z^t) \odot (1 - m) + \phi(z_e^t) \odot m. \tag{5}$$

**Spatially Propagated Editing.** To enhance the quality of reconstructed 4D scenes on monocular videos that may lack multi-view supervision, potentially leading to artifacts or poor structures in the 4D scene, we implement a random pose interpolation strategy for the progressive scene reconstruction process. This interpolation involves randomly generating a novel pose between two poses in their corresponding frames, $E'_{i_1}$ and $E'_{i_2}$, along the original camera trajectory. Specifically, we utilize $\text{Slerp}(P_{i_1}, P_{i_2}, \theta)$ to represent the spherical linear interpolation between the corresponding poses $P_{i_1}$ and $P_{i_2}$ from $E_{i_1}$ and $E_{i_2}$, respectively, with the interpolation coefficient $\theta$.

To ensure that the interpolated pose can be rendered consistently with the edited frames from TPE, we use the known view as a reference to refine the novel view from an interpolated pose. Specifically, we extract the latent token of the novel view $E'_{novel}$ rendered with the pose $P_{novel}$ at the same timestamp as $E'_{i_2}$. We then obtain the attention features with the corresponding latent token extracted from $E'_{i_2}$ using $\hat{p}_o$ following Eq. 1 and Eq. 2. Utilizing a technique similar to TPE, we propagate the attention feature from $E'_{i_2}$ to $E'_{novel}$ as described in Eq. 3 to refine the edited image $\hat{E}'_{novel}$ based on the novel pose.

Subsequently, we further refine the rendered image of the novel view to enhance the 4D scene reconstruction by calculating the loss $\mathcal{L}_{novel}$ using the L1 loss function:

$$\mathcal{L}_{novel} = \mathcal{L}_1(E'_{novel}, \hat{E}'_{novel}). \tag{6}$$

## 3.2 Progressive 4D Gaussian Splatting

In 3D or 4D editing tasks, it is tedious for users to compute the camera poses with SfM libraries such as COLMAP (Schonberger & Frahm, 2016). Moreover, COLMAP may not be able to accurately estimate poses from various types of casual videos, particularly those featuring dynamic scenes with highly dynamic objects, poorly textured surfaces, and rotating camera motions that make it challenging to match features. Thus, we propose an efficient 4D scene representation that eliminates the need for users to compute camera poses, accommodates casual monocular videos as input, and streamlines the entire editing process.

**Relative Pose Estimation.** Inspired by CFGS (Fu et al., 2024), we introduce the local 3DGS $G_i^l$ to estimate the relative camera pose between two consecutive frames, which is then utilized as input for the 4DGS pose. This technique involves estimating the relative camera pose by applying a learnable SE-3 affine transformation $P_i$ to the 3D Gaussian $\hat{G}_i^l$ reconstructed from the current frame $n$ to obtain the 3D Gaussian representation for the subsequent frame $i + 1$, denoted as $G_{i+1}^l = P_i \odot G_i^l$. The transformation $P_i$ is optimized by minimizing the photometric loss between the rendered image and the current edited frame $E_{i+1}$.

$$\hat{P}_i = \arg\min_{P_i} \mathcal{L}_{rgb}(\mathcal{R}(P_i \odot G_i^l), E_{i+1}), \tag{7}$$

where $\mathcal{R}$ is the rendering process for the local 3DGS and the 4DGS. Note that the attributes of $\hat{G}_i^l$ are fixed to differentiate camera motion from other Gaussian transformations including pruning, densification, and self-rotation.

**4D Gaussian Representation.** We develop a 4D scene representation aimed at enhancing editing efficiency and reconstruction performance. As illustrated in Fig. 2, we design temporal components to capture time-aware motion and deformation. In particular, we introduce an attribute known as the *appearing time* ($\tau$), specifying from when each point actively contributes within the 4D Gaussians. By combining this temporal attribute with time-dependent functions, we can effectively model the parameters of the 4D Gaussians, *i.e.*, center position, opacity, scale, and rotation. This approach enables us to accurately represent scene content that emerges or disappears over the duration of the video.

For each Gaussian at time $t_g$, we utilize a time-dependent function to model its motion. We select the polynomial function to represent the current position at time $t_g$, denoted as $\mu_i(t)$:

$$\mu_i(t) = \sum_{k=0}^{n} b_{i,k}(t_g - \tau_i)^k, \tag{8}$$

where $\tau_i$ denotes the appearing time of each Gaussian. We choose $n = 3$ for the 3rd-degree polynomial function for a balance between model size and performance. The coefficients $b_{i,k} \in \mathbb{R}$ associated with this function are optimized during training. Similarly, for rotational motion, we

utilize the 1-st polynomial functions with $n = 1$ to represent the rotation $q_i(t)$:

$$q_i(t) = \sum_{k=0}^{n} c_{i,k}(t_g - \tau_i)^k, \tag{9}$$

where $c_{i,k} \in \mathbb{R}$ are the polynomial coefficients, optimized during training.

Furthermore, we employ a temporal radial basis function to describe the temporal opacity $o_i(t)$:

$$o_i(t) = o_i^s \exp(-s_i^{\hat{\tau}} \|t_g - \tau_i\|^2), \tag{10}$$

where $s_i^{\hat{\tau}}$ is a temporal scaling factor optimized during training, $\hat{\tau} = t_g - \tau_i$, and $o_i^s$ denotes the time-independent spatial opacity.

**Progressive 4D reconstruction.** To begin the training process, we utilize a pre-trained monocular depth estimator (Ranftl et al., 2021) to generate the depth map $D_i$ from the edited frame $E_i$. This depth map offers robust geometric information independent of camera parameters. We initialize our 4D representation with points lifted from the monocular depth using camera intrinsic and orthogonal projections. Subsequently, we train the 4D Gaussians with all attributes by minimizing the photometric loss and depth loss between the rendered image and the current edited frame $E_n$:

$$G_n^* = \arg\min_{G_n}(\lambda_r \mathcal{L}_{rgb}(\mathcal{R}(G_n), E_n), \lambda_d \mathcal{L}_{dep}(\mathcal{R}(G_n), D_n)), \tag{11}$$

where $\lambda_r$ and $\lambda_d$ are the coefficients for photometric loss and depth loss respectively.

The photometric loss $\mathcal{L}_{rgb}$ is $\mathcal{L}_1$ combined with a D-SSIM loss:

$$\mathcal{L}_{rgb} = (1 - \lambda)\mathcal{L}_1 + \lambda \mathcal{L}_{D-SSIM}, \tag{12}$$

where $\lambda = 0.2$ is empirically set for all experiments. The depth loss can be represented as:

$$\mathcal{L}_{dep} = \mathcal{L}_1(\mathcal{R}(G_n), D_n) \tag{13}$$

During the progressive training process, we calculate the relative camera pose between consecutive frames using the local 3DGS, which serves as the initial camera pose for our 4DGS. Subsequently, the 4DGS updates the set of 4D Gaussians with all attributes based on the learnable camera poses $P_i$ obtained from the local 3DGS. Recognizing that accumulated pose errors from the local 3DGS estimations could impede the optimization of a global scene, we iteratively update the rendered images and camera poses within observed frames. Additionally, we interpolate the novel pose between the observed frames using SPE, and use the refined results of novel views from SPE to supervise the editing of the original novel views, as outlined in Eq. 6. We provide the pseudocode for our 4DEditPro in the Appendix.

## 4 EXPERIMENTS

### 4.1 EXPERIMENTAL SETUP

**Datasets.** We reconstruct and edit 4D scenes from three public datasets: 1) DAVIS (Perazzi et al., 2016), which consists of monocular videos without camera poses, 2) Tanks & Templates (Knapitsch et al., 2017), featuring complex camera pose movements, and 3) SemanticKITTI (Behley et al., 2019), showcasing complex and large driving scenes. Our method introduces a direct editing process that eliminates the need for extracting camera poses from COLMAP or initializing 4D scene representation. However, for comparison purposes, we use COLMAP to extract camera poses in Tanks & Templates and SemanticKITTI datasets for the approaches under comparison (Haque et al., 2023; Chen et al., 2024), and utilize the pose estimated in DAVIS datasets by our method as the initial pose for the other methods. Additionally, we evaluate our method on causal monocular videos captured by an iPhone, as showcased in the Appendix.

**Evaluation Metrics.** The quality of 4D scene editing can be assessed on fidelity and temporal consistency. Following common practice, we calculate the average similarity between the CLIP embedding of each edited frame and the target text prompt (Radford et al., 2021) as the CLIP score, as well as the mean SSIM score between the rendered edited frames warped by optical flow (Teed & Deng, 2020) and the corresponding original frames as WarpSSIM (Shin et al., 2024). In addition to these evaluation metrics, we conduct a user study to assess the quality of 4D scene editing on the DAVIS and Tanks & Templates datasets. This user study involves a two-way or three-way voting process to compare our method with other state-of-the-art approaches.

Table 1: Quantitative results on the DAVIS (Perazzi et al., 2016) with different scenes. * denotes that the scores are calculated solely for the regions of dynamic objects.

| Scene | Instruction | CLIP score↑ | | WarpSSIM↑ | |
|---|---|---|---|---|---|
| | | GSEditor-4D | **Ours** | GSEditor-4D | **Ours** |
| Black Swan | "Origami" | 0.2695 | 0.2886 | 0.7763 | 0.7818 |
| | "Van Gogh"* | 0.1982 | 0.2008 | 0.7682 | 0.9388 |
| Rhino | "Silver" | 0.2119 | 0.2615 | 0.7455 | 0.9098 |
| | "Night" | 0.1388 | 0.1449 | 0.8180 | 0.8328 |
| Boat | "Steampunk" | 0.1762 | 0.1829 | 0.8096 | 0.8598 |
| | "Oil painting" | 0.2128 | 0.2398 | 0.7352 | 0.7387 |
| Average | | 0.2012 | **0.2198** | 0.7755 | **0.8436** |

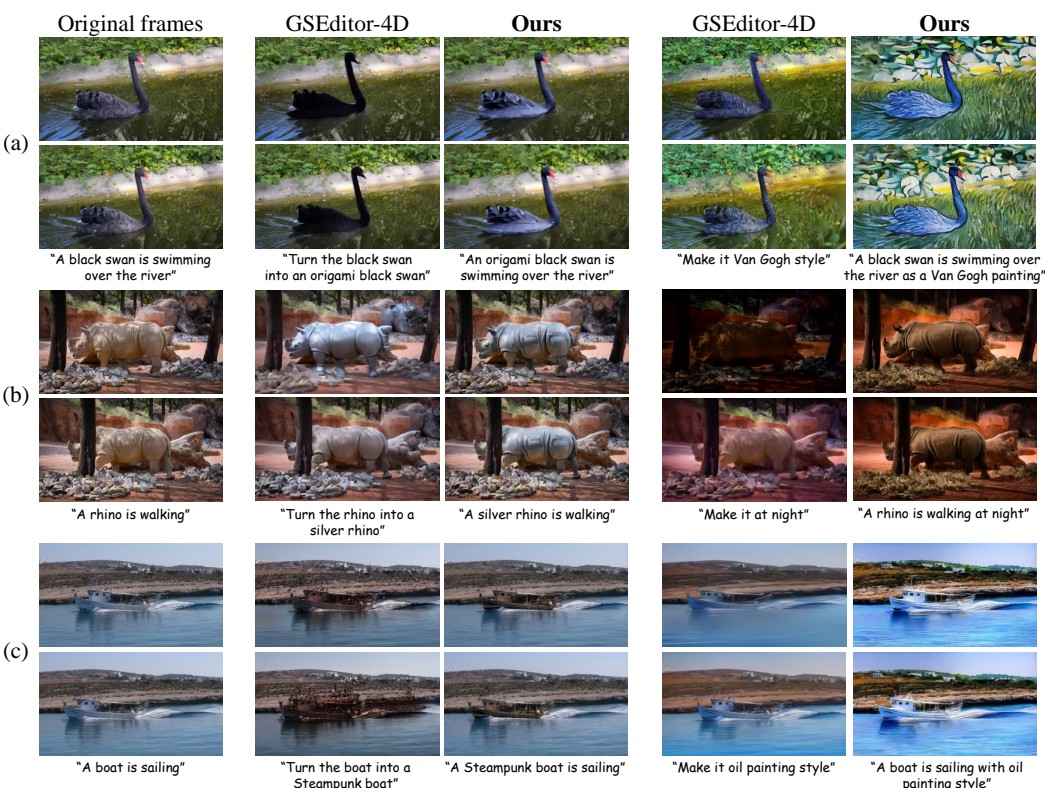

Figure 4: Qualitative results on DAVIS (Perazzi et al., 2016) datasets. **Better viewed when zoomed in.**

**Implementation Details.** Our method is implemented using the PyTorch library (Paszke et al., 2017). We use StableDiffusion v2.1 from the Hugging Face library as our main diffusion model for editing the scene. we set the classifier-free guidance scale of the TPE module at around 7.5 and that of the SPE module at 1.5 for best performance. To generate masks for local editing, we utilize Lang SAM (Kirillov et al., 2023) based on the local segmentation prompt. Typically, the complete editing process for our project takes about 25 minutes to handle 40-50 frames of a scene. Notably, this process does not need COLMAP precomputation or model initialization and can be executed efficiently on a single NVIDIA 48GB L20 GPU. Further details on implementation are available in the Appendix.

## 4.2 METHODS UNDER COMPARISON

To the best of our knowledge, our 4DEditPro is the first method of editing 4D scenes from casual monocular videos without the need for camera pose input, whereas other 4D editing methods (Shao et al., 2023; Mou et al., 2024) require camera pose input. Since the source code of these methods have not been released publicly, we develop GSEditor-4D based on GaussianEditor (Chen et al.,

Table 2: Quantitative results on Tanks & Templates (Knapitsch et al., 2017) and SemanticKITTI (Behley et al., 2019) datasets with different scenes.

| Dataset | Scene | Instruction | CLIP score↑ | | | WarpSSIM↑ | | |
|---|---|---|---|---|---|---|---|---|
| | | | IN2N | GSEditor | **Ours** | IN2N | GSEditor | **Ours** |
| Tanks & Templates | Horse | *"Brown horse"* | 0.2145 | 0.1864 | 0.2179 | 0.9025 | 0.8735 | 0.9107 |
| | | *"Snowy"* | 0.1815 | 0.1329 | 0.2013 | 0.8613 | 0.8656 | 0.9024 |
| | Ignatius | *"Sand"* | 0.1620 | 0.1864 | 0.1909 | 0.5859 | 0.9101 | 0.8970 |
| | | *"Minecraft"* | 0.2089 | 0.2195 | 0.2399 | 0.8619 | 0.8243 | 0.8659 |
| SemanticKITTI | Driving | *"Railway"* | 0.1595 | 0.1721 | 0.2170 | 0.1665 | 0.1967 | 0.7571 |
| | | *"Autumn"* | 0.1860 | 0.1824 | 0.2301 | 0.2369 | 0.1310 | 0.7655 |
| Average | | | 0.1854 | 0.1845 | **0.2162** | 0.6025 | 0.6335 | **0.8498** |

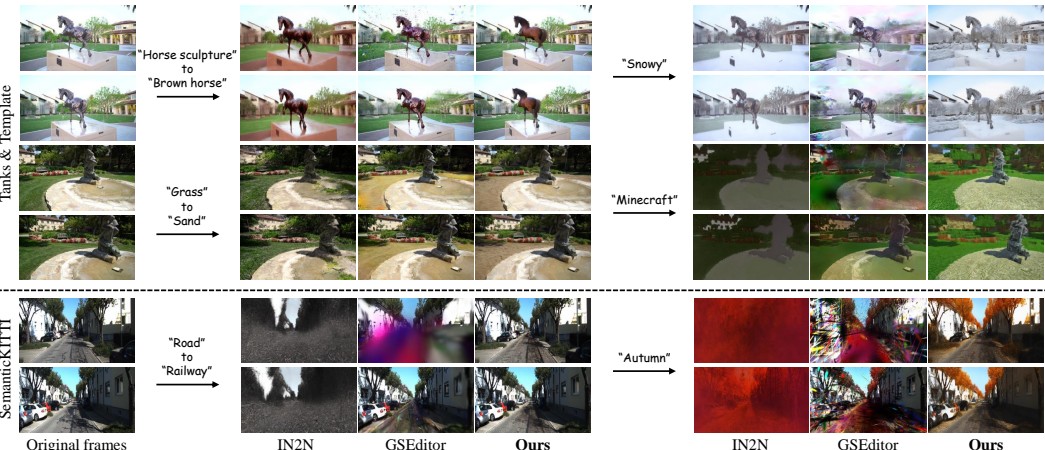

Figure 5: Qualitative results on Tanks & Templates (Knapitsch et al., 2017) and SemanticKITTI (Behley et al., 2019) datasets. **Better viewed when zoomed in.**

2024), extending its 3D Gaussian representation to a 4D representation and utilizing our estimated camera poses as its input.

For the DAVIS (Perazzi et al., 2016) dataset, we have conducted qualitative and quantitative comparisons between our method and GSEditor-4D. Since both Tanks & Templates (Knapitsch et al., 2017) and SemanticKITTI (Behley et al., 2019) datasets contain multi-view images, we not only compare with GaussianEditor (GSEditor) (Chen et al., 2024) but also Instruct-NeRF2NeRF (IN2N) (Haque et al., 2023) on these two datasets.

### 4.3 RESULTS

To compare the global and local editing performance on 4D scenes among different methods, we select various text prompts to test.

**4D Editing Results on DAVIS Datasets.** The quantitative results in Tab. 1 show that our method achieves high scores in CLIP score and WarpSSIM, indicating better editing fidelity and temporal consistency than GSEditor-4D. Qualitative results are presented in Fig. 4, where, for example, the target prompt "make it at night" for GSEditor-4D displays varying and uneven illumination across different frames. In contrast, our approach maintains consistent illumination and appearance. These results demonstrate the effectiveness of our method in both local and global scene editing. Further results are available in the Appendix.

**4D Editing Results on Other Datasets.** In Tab. 2, we present three complex scenes with two text prompts, one for global and the other local editing, for each scene, as examples to compare the CLIP score and WarpSSIM metrics. Complemented by the qualitative results in Fig. 5, our method has been shown to outperform previous approaches on producing consistent 4D scenes accurate to the prompt, especially in the driving scene from the SemanticKITTI dataset. To showcase the 4D attributes of our reconstructed scenes, we include the rendered depth maps corresponding to the rendered images in the Appendix.

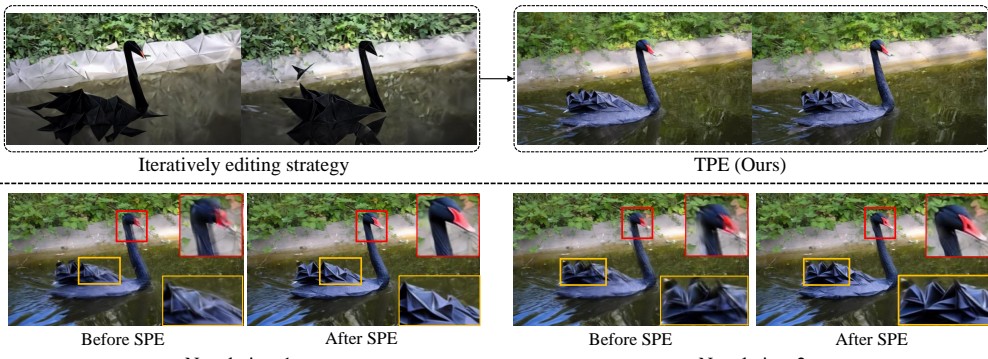

Figure 6: Qualitative ablation study on DAVIS datasets. **Better viewed when zoomed in.**

**User Preference Study.** To further assess qualitative results, we conducted experiments on 10 different text instructions using DAVIS datasets and 9 different text prompts using Tanks & Templates datasets. We used two- and three-alternative forced-choice decision methods to ask 21 users to choose their preferred results based on the relevance of the edited images or videos with text prompts, visual quality, and consistency. As shown in Tab. 3, our 4DEditPro was generally preferred over other methods on both datasets.

### 4.4 ABLATION STUDY

We conducted an ablation study on the "black swan" scene from the DAVIS dataset (Perazzi et al., 2016) with the text prompt: "An origami black swan is swimming over the river". First, we replaced TPE by directly utilizing latent tokens from DDIM inversion and denoising all tokens based on the text prompt without propagating reference attention features (w/o TPE). Second, we omitted the use of SPE in our methodology by excluding the SPE process from our pipeline (w/o SPE). Third, we investigated the impact of local 3DGS on our model by directly learning the camera pose parameters within the 4D Gaussian representation (w/o local 3DGS). Tab. 4 presents the quantitative outcomes of our ablation studies. It shows TPE's impact is the most significant whereas both SPE and local 3DGS enhance WarpSSIM only. To delve deeper, we provide an additional ablation study to assess the significance of local 3DGS, which crucially contributes to the learning of camera poses essential for scene reconstruction. This analysis is detailed in the Appendix.

Table 3: The user preference study on DAVIS and Tanks & Templates datasets.

| Dataset | Method | User Study (%) |
|---|---|---|
| DAVIS | GSEditor-4D | 24.50% |
| | Ours | **75.50%** |
| Tanks & Templates | IN2N | 17.22% |
| | GSEditor | 13.89% |
| | Ours | **68.89%** |

Also, we present a qualitative ablation study depicted in Fig. 6. We compare with the previous iteratively per-image editing strategy applied to 4D scene editing on the "black swan" scene from the DAVIS dataset. The results clearly show temporal inconsistencies in the adjacent frames when using the previous strategy, whereas our TPE module produces results of higher consistency, as shown in the first row of the figure. Furthermore, the second row of the figure displays the rendered results of the novel view before and after applying the SPE, which demonstrates that the use of SPE refines details and reduces artifacts in the novel view scenes.

Table 4: The ablation study of each component.

| Modules | CLIP score ↑ | WarpSSIM ↑ |
|---|---|---|
| ALL | **0.2886** | **0.7818** |
| w/o TPE | 0.2641 | 0.6799 |
| w/o SPE | 0.2823 | 0.7619 |
| w/o local 3DGS | 0.2863 | 0.7297 |

## 5 CONCLUSION

This paper introduces a novel 4D editing framework, named 4DEditPro, which utilizes casual monocular video input along with text prompts for editing. We introduce TPE and SPE modules to aid the diffusion model in producing 4D consistent editing results in both temporal and spatial aspects. Furthermore, we have developed a progressive 4D Gaussian Splatting pipeline to effectively reconstruct the edited 4D scene while estimating the camera pose. Through extensive experiments on three public datasets with multiple evaluation metrics, we demonstrate the effectiveness of our method.

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

# A  APPENDIX

## A.1  MORE IMPLEMENTATION DETAILS

**Details for Diffusion Models.** We employ DDIM inversion with a classifier-free guidance scale of 1, consistent with TokenFlow (Geyer et al., 2024). We employ DDIM deterministic sampling with 50 steps in TPE and 20 steps in SPE. We downsample 2x of the input sequences or images in TPE and SPE to reduce the computational cost and improve the editing speed.

**Details for Temporally Propagated Editing (TPE).** In the TPE module, we begin by partitioning the entire video sequence into multiple batches. Within each batch, we select $K_b$ frames, where $K_b$ is the largest integer between 1 and 10 that can evenly divide the total number $N_v$ of frames. After determining the number of batches as $N_B = \frac{N_v}{K_b}$, we proceed to randomly select one reference token from each batch. This selection yields $N_k$ reference tokens, where $N_k = N_B$, and $N_p$ remaining frames, with $N_p = N_v - N_k$.

**Details for Related Pose Estimation.** In pose estimation, we introduce local 3DGS to analyze pose changes between consecutive frames. However, in dynamic scenarios, the motion of some dynamic objects can hinder precise pose estimation in local 3DGS, particularly when movements are significant. To address this problem, we obtain dynamic object masks $M_d$ to exclusively consider static regions for more accurate pose estimation. Specifically, we first estimate the fundamental matrix using optical flow (Teed & Deng, 2020) between consecutive frames. Subsequently, by computing the threshold for the Sampson distance with the epipolar line, we obtain dynamic object masks $M_d$ to refine the accuracy of pose estimation in 4D dynamic scenes. We provide the pseudocode for our 4DEditPro in Alg. 1.

## A.2  MORE RESULTS

### A.2.1  MORE 4D EDITING RESULTS

To further evaluate the effectiveness of our methods in various scenarios, we present additional qualitative results using the DAVIS dataset (Perazzi et al., 2016) and the Tanks & Templates dataset (Knapitsch et al., 2017). In Fig. 9, we show five new scenes with diverse text prompts, including foreground object manipulation, background scenario modifications, and global scene style adjustments. Additionally, we display the rendered depth outcomes from the modified 4D scene, showing the geometric coherence of our results.

In Fig. 10, we show five editing scenarios of different text prompts. The consecutive rendered frames highlight the temporal consistency of our editing method, suggesting the effectiveness of our approach. The editing results in the last row of Fig. 9 and Fig. 10 do not utilize a mask in the

Table 5: The novel view synthesis results across 20 scenes from DAVIS datasets (Perazzi et al., 2016).

| Method | PSNR ↑ | SSIM ↑ | LPIPS ↓ |
|---|---|---|---|
| RoDynRF (Liu et al., 2023) | 24.67 | 0.6818 | 0.3963 |
| CFGS (Fu et al., 2024) | 24.18 | 0.7974 | 0.2363 |
| **Ours** | **30.18** | **0.8818** | **0.1453** |

TPE module. Further discussion on this can be found in Sec. A.3.2. Overall, these results collectively demonstrate the effectiveness of our methodology in diverse scenarios.

In addition, we demonstrate the scene edited using our 4DEditPro on casual monocular videos captured by an iPhone in Fig. 8.

---

**Algorithm 1:** 4DEditPro

---

1   **Input:** $\hat{I} = [I_1, I_2, ..., I_n], \hat{p}_v, \hat{p}_e, \hat{p}_o$;         $\triangleright$ Input Video, Text Prompts

2   $\hat{\phi}(z) = [\phi(z_1), \phi(z_2), ..., \phi(z_n)] \leftarrow$ DDIMInversion$(\hat{I}, \hat{p}_v)$;

3   $\hat{E} = [E_1, E_2, ..., E_n] \leftarrow$ TPE$(\hat{I}, \hat{p}_e, \hat{\phi}(z))$ ;

4   $i \leftarrow 0$ ;

5   **while** $i < n$ **do**

6      $D_i \leftarrow$ DepthEstimator$(E_i)$ ;

7      $\mu_i \leftarrow$ Initialization$(D_i)$ ;             $\triangleright$ Initialize 3D Position

8      **if** $i > 0$ **then**

9        $c, o, \Sigma, P_{i-1} \leftarrow$ InitAttributes$(G_i^l)$ ;     $\triangleright$ Colors, Opacities, Covariances, Camera Poses

10      **else**

11        $P_0 \leftarrow$ Random Initialization ;

12        $c, o, \Sigma, P_0 \leftarrow$ InitAttributes$(G_i^l)$ ;

13      **end**

14      $P_i \leftarrow \arg\min \mathcal{L}_{rgb}(\mathcal{R}(P_i \odot G_i^l), E_i)$ ;

15      $c_i, o_i, \Sigma_i, P_i, t \leftarrow$ InitAttributes$(G_i)$;   $\triangleright$ Colors, Opacities, Covariances, Camera Pose, Time

16      $E_i', D_i' \leftarrow \mathcal{R}(G_i)$ ;

17      **if** $i > 0$ **then**

18        $P_{novel} \leftarrow$ Slerp$(P_i, P_{prev}, \theta)$;          $\triangleright$ Spherical Linear Interpolation

19        $E_{novel}' \leftarrow \mathcal{R}(G_i(P_{novel}))$ ;

20        $\phi(z_i), \phi(z_{novel}) \leftarrow$ DDIMInversion $(E_{novel}', E_i', \hat{p}_o)$ ;

21        $\hat{E}_{novel}' \leftarrow$ SPE$(E_{novel}', E_i', \hat{p}_e, \phi(z_i), \phi(z_{novel}))$ ;

22      **end**

23      $\mathcal{L} \leftarrow \mathcal{L}_{rgb} + \mathcal{L}_{dep} + \mathcal{L}_{novel}$;            $\triangleright$ Total Loss

24      $G_i \leftarrow$ Adam$(\nabla \mathcal{L})$ ;           $\triangleright$ Back Propagation and Step

25      Pruning$(G_i)$ and CloneAndSplit$(G_i)$;       $\triangleright$ Pruning and Densification

26      $i \leftarrow i + 1$;           $\triangleright$ Progressively Reconstruction

27   **end**

---

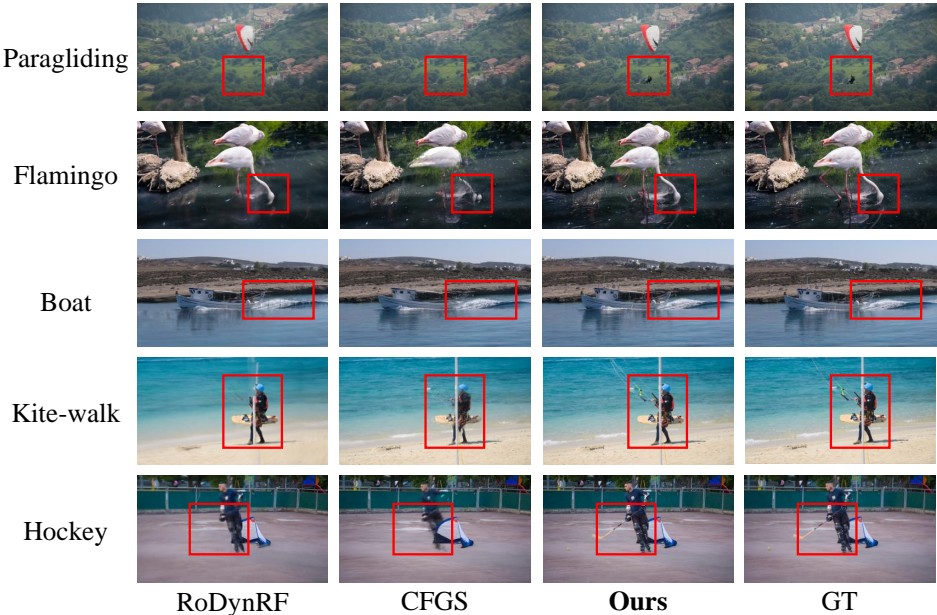

Figure 7: Comparison of novel view synthesis results on the DAVIS dataset. **Better viewed when zoomed in.**

### A.2.2   NOVEL VIEW SYNTHESIS RESULTS

We present novel view synthesis results to further assess our 4D Gaussian representation and pose estimation strategy in Sec. 3.2 compared to previous pose-free reconstruction methods, namely Ro-

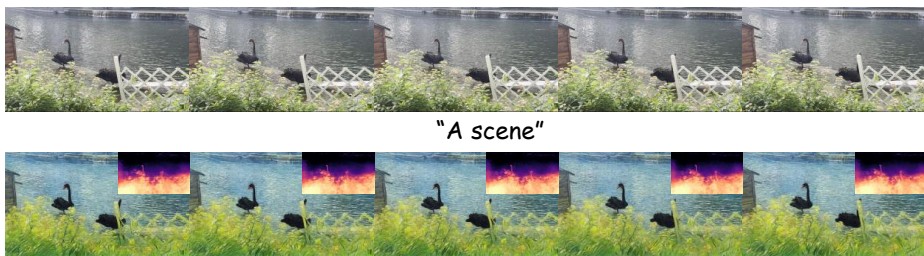

"A scene"

"A scene as a Van Gogh painting"

Figure 8: Editing results with the casual video captured by iPhone. **Better viewed when zoomed in.**

DynRF (Liu et al., 2023) and CFGS (Fu et al., 2024). Our evaluation covers 20 scenes from the DAVIS (Perazzi et al., 2016) dataset, comprising temporal monocular videos without camera pose input, with frames sampled at 1 per 8 frames.

Quantitative results in Tab. 5 show that our method outperforms the other two methods significantly in terms of PSNR, SSIM, and LPIPS scores. Moreover, as depicted in Fig. 7, the other methods struggle with reconstructing videos featuring significant movements and complex dynamic changes, *e.g.* paragliding and hockey scenes, often resulting in more artifacts. In contrast, our 4D representation excels in producing accurate and high-quality novel view synthesis results.

Table 6: Ablation studies with novel view synthesis results on each component on DAVIS (Perazzi et al., 2016) dataset.

| Method | PSNR ↑ | SSIM ↑ | LPIPS ↓ |
|---|---|---|---|
| All | **23.13** | **0.5991** | **0.2919** |
| w/o SPE | 22.31 | 0.5537 | 0.3212 |
| w/o local 3DGS | 21.57 | 0.5313 | 0.3484 |

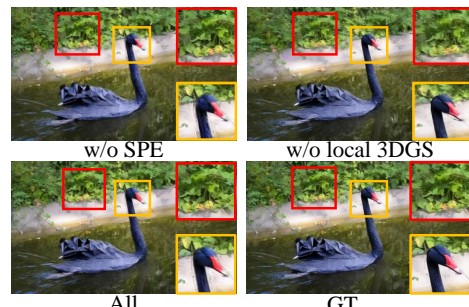

w/o SPE        w/o local 3DGS

All        GT

Figure 11: Visualization of Ablations on each component.

### A.2.3 DEMO VIDEO

We provide a demo video in our Supplementary Material showcasing the video results on various datasets with scene reconstruction, view interpolation, and some comparison.

### A.3 MORE ABLATIONS

### A.3.1 VISUALIZATION OF ABLATIONS ON EACH COMPONENT

To further evaluate the efficacy of our component, we present ablation results on novel view synthesis. We pick the task of novel view synthesis for this evaluation as it effectively demonstrates the spatial characteristics in views that lie beyond the original camera trajectory. The absence of TPE reduces the coherency in reconstructed results, leading to incomplete and incongruent images. Utilizing our TPE module is important for the performance, because it leverages the propagation of attention features, in contrast to the reliance on the convergence of scene-based iteratively per-image editing or 3D warping methods that depend on the accuracy of optical flow.

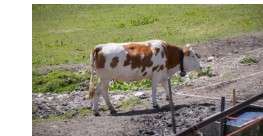

original frames

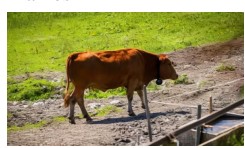

w/ masks        w/o masks

Figure 12: Visualization of Ablations on mask for local edits.

In Tab. 6 and Fig. 11, the impact of omitting local 3DGS is evident: the reconstructed background appears blurry not only within the yellow box but also in the red box. Similarly, the absence of SPE results in increased artifacts in the novel view. The quantitative results also demonstrate the significance of both the SPE and local 3DGS components. These results are averaged across five novel views sampled from the edited video sequences. Consequently, when all components are utilized, the results exhibit enhanced spatial consistency.

### A.3.2 MASKS ON LOCAL EDITING

To achieve a balance between precise local editing and the effectiveness of global editing, we incorporate the masks extracted by Lang SAM (Kirillov et al., 2023), which indicate the areas that need TPE, as in Eq. 5.

As shown in Fig. 12, when a mask is employed to direct the editing process, changes mainly occur in the masked regions and regions outside the mask remain consistent with the original frames. In comparison, in the absence of a mask, in addition to the target areas of the editing prompt, other areas may undergo minor alterations in color or contrast. This observation offers users the flexibility to choose whether to utilize a mask as a condition for the editing process.

### A.4 DETAILS OF USER PREFERENCE STUDY

In our user preference study, participants were presented with a series of questions, each featuring an original view or video along with rendered views or videos from various methods. An example question is shown in Fig. 13, where participants were asked to select their preferred rendered image or video. To ensure unbiased responses, the order of the methods was randomly set for each question, and all options were presented anonymously.

### A.5 LIMITATIONS

Our 4DEditPro framework effectively edits 4D scenes from casual monocular videos without requiring camera pose input or model initialization. Nevertheless, it has several limitations. Firstly, the 4D Gaussian representation in our 4DEditPro is developed by combining time-dependent functions or parameters to learn the attributes of Gaussians, which may struggle to fit the scene with complex motions. Introducing a more sophisticated modeling method, such as using Multilayer Perception (MLP) to learn dynamic attributes, may be necessary. Another issue is that 4DEditPro depends on the generative and editing abilities of the base diffusion model. Editing efficacy may be compromised if the base model is unable to handle certain editing tasks effectively.

**Discussion on DDIM Inversion.** In editing tasks, we incorporate DDIM inversion to extract latent tokens, enabling the acquisition of consistent initial noise that aligns with temporal and spatial editing outcomes for coherence. However, during the process of inverting the original video to latent tokens, additional uncontrolled noise might be introduced. This noise can manifest as subtle disturbances when reconstructing these latent tokens back to recreate the original video. Fine-tuning strategies may be applied to constrain and mitigate such noise artifacts.

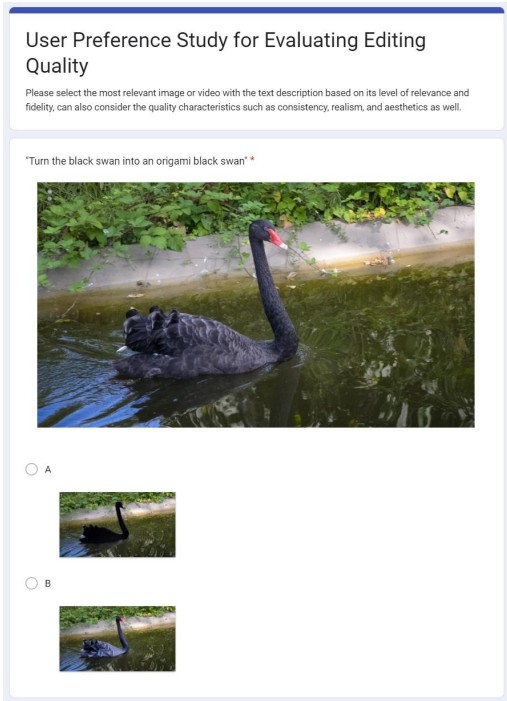

Figure 13: An example of our user preference study.

**Discussion on Progressive 4D Gaussian Splatting.** In the progressive reconstruction of 4D edited scenes, drift errors may arise from estimating long video sequences using local 3DGS. To mitigate this problem, keyframe selection can be implemented to detect and optimize the learnable camera pose when progressively learning video sequences of a certain length. Furthermore, we are exploring pose estimation methods of higher robustness to enhance scene reconstruction accuracy.

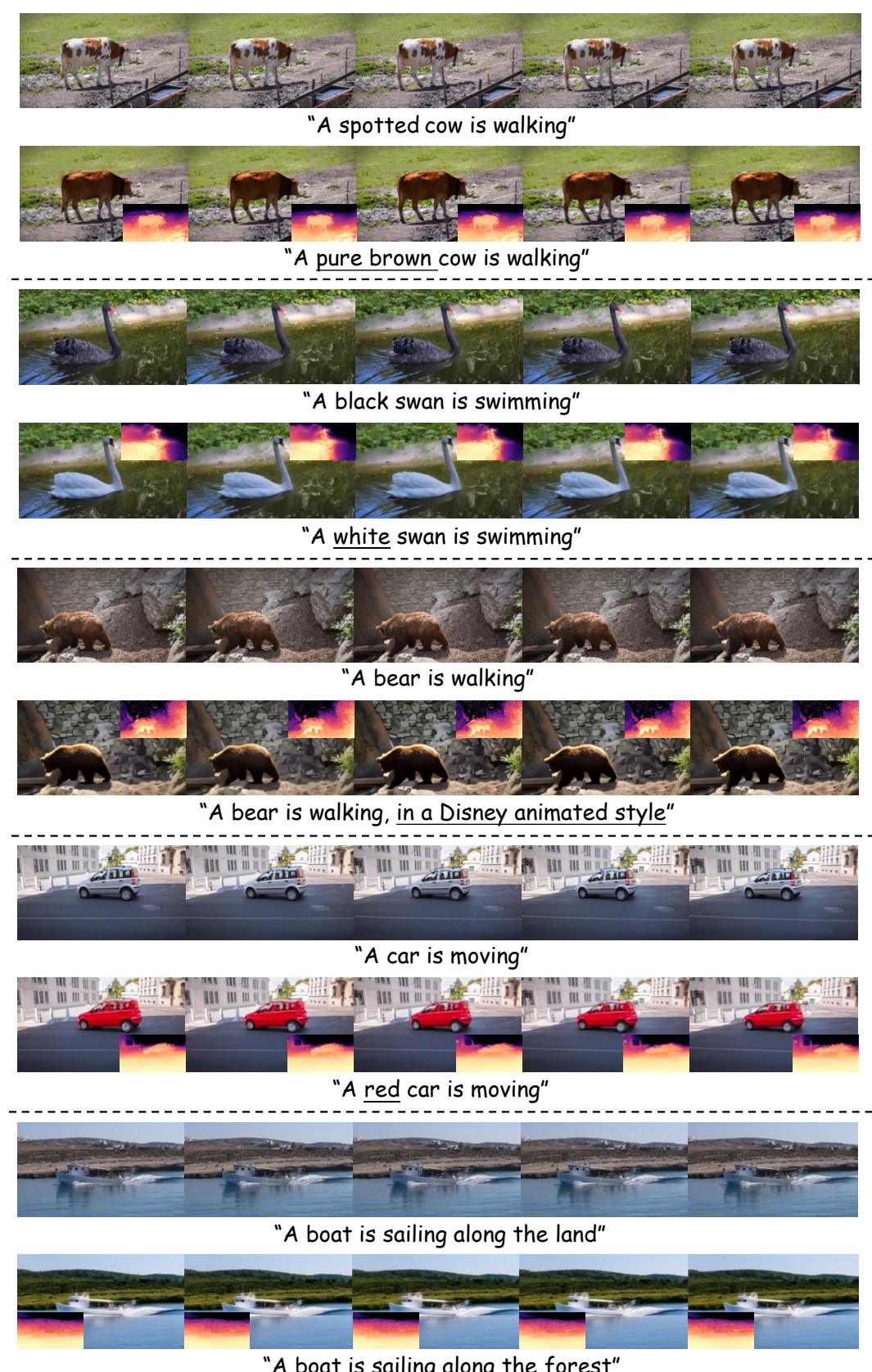

Figure 9: Additional editing results are provided on the DAVIS dataset (Perazzi et al., 2016). For each scene, the first row shows the original video frames, while the second row displays the rendered frames and depth of the reconstructed 4D edited scenes. **Better viewed when zoomed in.**

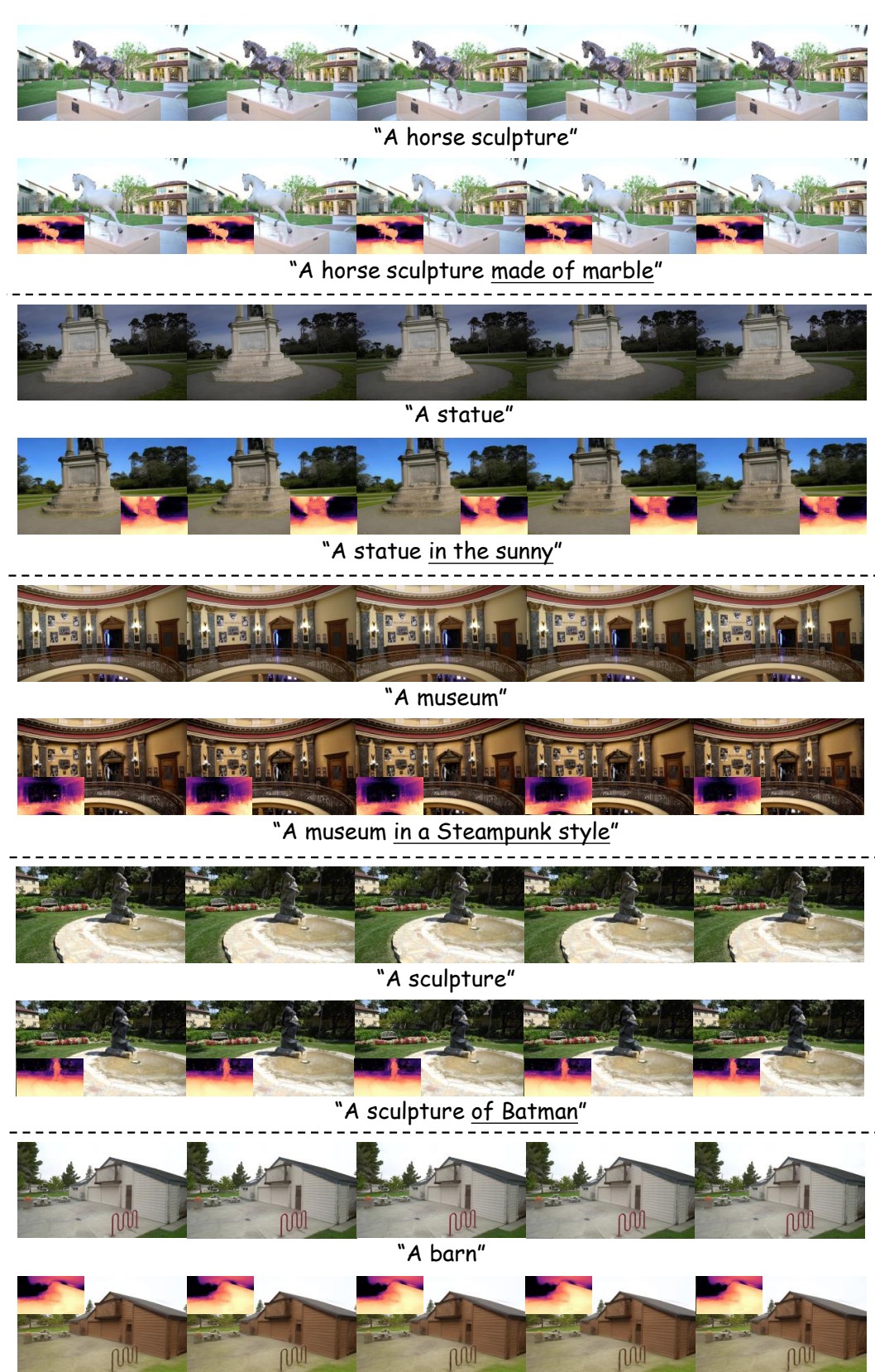

Figure 10: Additional editing results are provided on the Tanks & Templates dataset (Knapitsch et al., 2017). For each scene, the first row shows the original video frames, while the second row displays the rendered frames and depth of the reconstructed 4D edited scenes. **Better viewed when zoomed in.**

