# OpenReview forum: "4DEditPro: Progressively Editing 4D Scenes from Monocular Videos with Text Prompts"
_ICLR.cc/2025/Conference — ICLR 2025 Conference Withdrawn Submission_

### Official Review · Reviewer_jobR · 2024-10-31

**Soundness:** 3
**Presentation:** 3
**Contribution:** 2
**Rating:** 3
**Confidence:** 4

**Summary:**

The paper propose a framework for 4D scene editing in casual monocular video using text prompt. Unlike traditional methods that require multi-view images or camera poses, 4DEditPro works without external tools by using two key modules: TPE for maintaining coherence across frames and SPE for enhancing spatial consistency. A pose-free 4D Gaussian Splatting(4DGS) approach further enables scene reconstruction and editing without pre-calculated poses. Experiments demonstrate the result of this 4DEditPro through both qualitative and quantitative results, as well as user evaluations.

**Strengths:**

1. The paper is in general well organized and easy to follow.
2. The paper presents a pipeline for performing editing 4D from monocular video using Gaussian Splatting.

**Weaknesses:**

1. The key contribution of the proposed method (TPE, SPE) appears to lie in the integration of several minor techniques, such as feature extraction and injection into the video diffusion model. The novelty seems lacking, without too much novel insight.

2. The advantages of TPE are not clear. Comparative experiments with previous off-the-shelf video editing models(e.g., TokenFlow[1], Fatezero[2], Flatten[3]) would be essential to demonstrate TPE’s advantages. However, this paper only includes comparisons with 3D editing models like GSEditor-4D.

3. The pose changes in the novel view synthesis from the monocular video in the demo video appear too subtle. These result seems closer to video editing and falls somewhat short of being considered 4D editing. A detailed disclosure of how the authors set the poses in this experiment with monocular video would enhance the understanding of this paper’s strengths.

4. Additionally, while the paper claims to achieve 4D editing, no experiments exist on 4D datasets. Using representative 4D datasets like DyNeRF[4] and HyperNeRF[5], as well as comparisons with other 4D editing models (e.g.,Instruct 4D-to-4D[6]), would make the paper’s argument more persuasive.

[1] Geyer, Michal, et al. "Tokenflow: Consistent diffusion features for consistent video editing." arXiv preprint arXiv:2307.10373 (2023).
[2] Qi, Chenyang, et al. "Fatezero: Fusing attentions for zero-shot text-based video editing." Proceedings of the IEEE/CVF International Conference on Computer Vision. 2023.
[3] Cong, Yuren, et al. "Flatten: optical flow-guided attention for consistent text-to-video editing." arXiv preprint arXiv:2310.05922 (2023).
[4] Li, Tianye, et al. "Neural 3d video synthesis from multi-view video." Proceedings of the IEEE/CVF Conference on Computer Vision and Pattern Recognition. 2022.
[5] Park, Keunhong, et al. "Hypernerf: A higher-dimensional representation for topologically varying neural radiance fields." arXiv preprint arXiv:2106.13228 (2021).
[6] Mou, Linzhan, et al. "Instruct 4D-to-4D: Editing 4D Scenes as Pseudo-3D Scenes Using 2D Diffusion." Proceedings of the IEEE/CVF Conference on Computer Vision and Pattern Recognition. 2024.

**Questions:**

1. The paper claims to achieve 4D scene editing from monocular video, but isn’t this simply a combination of monocular video editing and 4D reconstruction from monocular video? I’m uncertain why this qualifies as 4D editing.

2. The explanation for setting the camera pose using Slerp is unclear. A more detailed clarification on this aspect would be helpful. Additionally, it would strengthen the paper to demonstrate how robustly the 4D reconstruction handles variations in camera poses.

---

### Official Review · Reviewer_GEno · 2024-11-04

**Soundness:** 1
**Presentation:** 3
**Contribution:** 2
**Rating:** 3
**Confidence:** 5

**Summary:**

This paper studies instruction-guided 4D scene editing. It proposes 4DEditPro, a method that uses two modules, TPE and SPE, to ensure the temporal and spatial consistency, and uses a pose-free 4DGS to reconstruct 4D scene from each viewpoint's videos. The proposed method can perform well in different 4D editing tasks in the evaluation.

**Strengths:**

- The paper is overall written clearly, with clear formulas and descriptions.
- In various editing tasks, the proposed method outperforms the baselines both quantitatively and qualitatively.
- Ablation studies are provided to show the effectiveness of components.

**Weaknesses:**

- The reasonability of task "4D scene editing using only casual videos" is questionable.
    - The 4D scene editing's input is a 4D _scene_, which is like a 3D model in Blender but is dynamic, that should be able to be put in any coordinates and render any videos accordingly.
    - The conversion between causal videos and 4D scenes is closer to 4D scene _reconstruction_ than editing. This should not be regarded as challenging in 4D editing.
    - Therefore, the challenges that this paper aims to solve are far-fetched - they are brought from another task (i.e., 4D scene reconstruction) to obtain the input of the current task (i.e. 4D scene editing), but not a part of the current task with a valid input.
    - In fact, lots of the contents of the paper just aim to do reconstruction, e.g., in Sec 3.2. This part seems quite orthogonal to the editing part.
- The model seems only working on monocular video, i.e. there is only one camera in the scene.
    - This significantly reduces the challenge of spatial 3D consistency. This might be the reason why a depth estimator (L335) can easily reconstruct the 3D structure.
    - When there is only one monocular video, the editing task then degrades to "video editing with 3D consistency requirements." Therefore, video editing methods should be compared. However, they are not.
    - According to the demo video, all the scenes are monocular. This necessitates the comparison against video editing models.
- The only baseline "Instruct 4D-to-4D" is not compared with. This is the only baseline that works in this task. It is crucial to compare with it.
    - The authors claimed that "Instruct 4D-to-4D"'s code is not publicly available. However, according to the Github repo of "Instruct 4D-to-4D", the code was released on 8/29, which is one month before the deadline of ICLR. This method should have, therefore, been compared against the paper.
    - Even if the code is not released, all the dataset used by Instruct 4D-to-4D are all public. Therefore, a comparison against Instruct 4D-to-4D should still have been achieved with those datasets and the same editing tasks as Instruct 4D-to-4D.
- In Tab.1, only the "Average" row is marked bold on the best numbers. Other rows should also be marked (and it seems that "Ours" are always the best, so this should improve the soundness).
- The model needs the user to provide descriptions of both original and edited scenes, which requires more human work. The baselines IN2N, GSEditor-4D, and Instruct 4D-to-4D only require editing instruction.
- The only 4D scenes used for editing are just three monocular dynamic scenes.
    - As a comparison, the baseline Instruct 4D-to-4D compares with at least 3 monocular dynamic scenes and 5 multi-view dynamic scenes, covering DyCheck, DyNeRF, Google Immersive, etc, and as long as 300 frames.
    - Therefore, this paper's comparison experiments are significantly weaker and more incomplete than the baseline.

**Questions:**

Following the weaknesses, please consider:
- Compare with video editing methods in spatial quality.
- Compare with baseline Instruct 4D-to-4D with its already released code.
- Compare with baseline Instruct 4D-to-4D by using the datasets it is using, i.e., DyCheck, DyNeRF, and Google Immersive, with the corresponding tasks.
- Evaluate the method on more multi-view scene datasets.

---

### Official Review · Reviewer_zW35 · 2024-11-04

**Soundness:** 2
**Presentation:** 2
**Contribution:** 2
**Rating:** 5
**Confidence:** 5

**Summary:**

The paper presents 4DEditPro, a new framework for editing 4D scenes in casual monocular videos using text prompts. Unlike conventional methods that require multi-view images or known camera poses, 4DEditPro works with single-view videos, allowing easy and consistent scene edits without extra setup. It achieves this by combining two modules: Temporally Propagated Editing (TPE) for smooth, time-consistent edits across frames. Spatially Propagated Editing (SPE) for spatial consistency by generating nearby “virtual views” to fill in missing details. Using a pose-free 4D Gaussian Splatting (4DGS) technique, 4DEditPro reconstructs scenes without needing camera poses, enabling flexible, high-quality editing. The approach is effective for both targeted edits and broader style changes, making text-driven video editing practical and seamless.

**Strengths:**

1. The task itself is valuable and timely, addressing the growing need for efficient 4D scene editing in casual videos.
2. The paper is well-structured and clearly written, making it easy for readers to understand the methodology and approach.
3. The experiments are thorough and appear to be well-designed, with enough detail provided to ensure reproducibility by others in the field.

**Weaknesses:**

1. The process for selecting the reference token lacks detail, and it’s unclear how this selection impacts the final results.
2. The pipeline doesn’t present any particularly innovative insights.
3. mThe editing results seem somewhat imprecise; for example, in Figure 4, the "silver" and "night" edits appear unnatural.

**Questions:**

Is there a dynamic demo available to assess the quality of the scene dynamics? I'd be interested in increasing my rating after seeing more extensive examples.

---

### Official Review · Reviewer_63VK · 2024-11-05

**Soundness:** 1
**Presentation:** 2
**Contribution:** 2
**Rating:** 3
**Confidence:** 3

**Summary:**

This paper introduces a framework for 4D scene editing from monocular videos guided by text prompts. The proposed techniques, Temporally Propagated Editing (TPE) and Spatially Propagated Editing (SPE), ensure temporal and spatial consistency in the editing process. By introducing progressive dynamic representation through 4DGS, the framework can model scene attributes without requiring camera pose as an input.

**Strengths:**

1. First approach for 4D scene editing from casual monocular videos, eliminating the need for camera pose input.
2. Introduced Temporally Propagated Editing (TPE) and Spatially Propagated Editing (SPE) modules to improve temporal and spatial consistency.
3. Quantitative evaluations show better performance over baselines, indicating the proposed method’s effectiveness across multiple metrics.

**Weaknesses:**

1. Temporal consistency is not maintained. In the supplementary video, noticeable flickering occurs in several segments, such as the sailing boat (00:26–00:28), Minecraft scene (00:45–00:47), horse editing (00:48–00:51), and statue editing (00:52–00:56).
2. The synthesized novel views show minimal differentiation from the original video, as seen in segments 00:22–00:23 and 1:08–1:12.
3. Furthermore, the supplementary video primarily demonstrates static view synthesis, despite the method being proposed for 4D editing.
4. The editing results showcased in the supplementary materials are mostly focused on color, style, and texture adjustments, with minimal instances of object shape editing. This suggests the method’s contributions in editing might be overstated.
5. In terms of comparisons, the paper primarily contrasts its approach with static 3D scene editing methods, even though it claims to support 4D editing. Given that the showcased editing focuses on color, style, and texture modifications, a more fitting baseline would involve applying a video style transfer technique to the input video, followed by reconstructing the 4D scene using methods designed for monocular videos.

**Questions:**

1. What factors contribute to the suboptimal performance of Gaussian Editor results? Given that scenes in Tanks and Temples and SemanticKITTI datasets are static (lacking moving objects), would it not be more appropriate to compare with the 3D version of Gaussian Editor? Furthermore, when applied to static scenes, do the results of the 3D and 4D versions of Gaussian Editor differ, or are they effectively the same?

---

### Note · Authors · 2024-11-13

I have read and agree with the venue's withdrawal policy on behalf of myself and my co-authors.